# Paternal dietary macronutrient balance and energy intake drive metabolic and behavioral differences among offspring

Angela Jane Crean [1], Alistair McNair Senior [1], Therese Freire[1], Thomas Daniel Clark[1], Flora Mackay[1], Gracie Austin[1], Tamara Jayne Pulpitel[1], Marcelo Aguiar Nobrega [2], Romain Barrès [3,4,5] ✉ & Stephen James Simpson [1,5] ✉

Paternal diet can influence the phenotype of the next generation, yet, the dietary components inducing specific responses in the offspring are not identified. Here, we use the Nutritional Geometry Framework to determine the effects of pre-conception paternal dietary macronutrient balance on offspring metabolic and behavioral traits in mice. Ten isocaloric diets varying in the relative proportion of protein, fats, and carbohydrates are fed to male mice prior to mating. Dams and offspring are fed standard chow and never exposed to treatment diets. Body fat in female offspring is positively associated with the paternal consumption of fat, while in male offspring, an anxiety-like phenotype is associated to paternal diets low in protein and high in carbohydrates. Our study uncovers that the nature and the magnitude of paternal effects are driven by interactions between macronutrient balance and energy intake and are not solely the result of over- or undernutrition.

Numerous reports have established that paternal diet (in addition to maternal diet) can influence offspring health and risk of disease[1–3]. Animal models have revealed that paternal diet affects offspring traits including metabolic and cardiovascular function[4,5], cancer risk[6], neurobiology and behavior[7,8], and reproductive health[9]. Considerable attention has been devoted to identifying different mechanisms of paternal effects, and understanding of the molecular and epigenetic signatures of sperm and seminal plasma has progressed rapidly with advances in sequencing capabilities[10–13]. However, the understanding and sophistication of nutritional models used in paternal effects studies have not kept pace with mechanistic knowledge.

Animal models of both paternal over- and under-nutrition show negative impacts on offspring metabolic health[3], suggesting paternal effects are not just a simple replication of phenotype across generations. In addition, offspring metabolism is influenced by many different types of environmental manipulations[14]. For example, paternal smoking[15], paternal restraint stress induced by immobilization for 2 h/day in a tube[16], paternal early postnatal stress induced by unpredictable separation from mothers[17], and removal of paternal seminal vesicles[18] have all been associated with changes in offspring metabolic traits. Given that diverse environmental manipulations induce similar changes in offspring, it is possible that increased rates of metabolic disease in offspring simply reflect paternal "environmental stress", and that future fathers should simply be advised to live a healthier, less stressful lifestyle. However, while generic stress may account for at least some of the changes observed in offspring[19], our understanding of paternal effects may also be enhanced by building more context-dependence and specificity into our theories and models[20].

Different dietary stress paradigms, such as high-fat diet, low-protein diet, or caloric restriction, have been used to uncover paternal

[1]Charles Perkins Centre and School of Life and Environmental Sciences, The University of Sydney, Sydney, NSW 2006, Australia. [2]Department of Human Genetics, University of Chicago, Chicago, IL 60637, USA. [3]Novo Nordisk Foundation Center for Basic Metabolic Research, University of Copenhagen, Copenhagen, DK 2200, Denmark. [4]Institut de Pharmacologie Moléculaire et Cellulaire, Université Côte d'Azur & Centre National pour la Recherche Scientifique (CNRS), Valbonne 06560, France. [5]These authors jointly supervised this work: Romain Barrès, Stephen James Simpson. ✉e-mail: barres@sund.ku.dk; stephen.simpson@sydney.edu.au

effects of epigenetic inheritance[21–23]. Yet, unavoidably, there are many differences between control and treatment diets in a two-diet design, and it is, therefore, unclear what difference (or combination of differences) in the paternal diet drives observed differences in offspring[24]. For example, are metabolic changes in offspring of fathers fed a 'Western' diet related to increased dietary fat, increased sugars, changed macronutrient ratios, increased calories, or some combination thereof? Are intergenerational effects of a 'low protein diet' related to decreased protein or commensurately increased carbohydrates and/or fats? Do dietary effects on palatability and appetite influence outcomes? Paternal effects are likely to be multi-dimensional (underpinned by many individual and interactive factors), with the importance of each factor varying in context-dependent ways[20]. However, if we want to develop dietary recommendations for males planning to conceive, it is essential to have a detailed and nuanced understanding of the direct and intergenerational consequences of each recommended dietary change.

As a first step in trying to unravel this complexity, we used the Nutritional Geometry Framework[25–27] in a mouse model to explore the effect of varying macronutrient balance in the paternal diet on offspring metabolic and behavioral traits. Diets were isocaloric and composed of identical ingredients. Metabolic and behavioral traits were measured at the same age in both fathers and their offspring, using the same methods. As food intake can be influenced by dietary composition (affecting calories consumed)[28,29], individual food intake of fathers was measured, and outcomes were assessed using both dietary proportions-based[30] and dietary intake-based[26] approaches.

This carefully designed and controlled study provided a new understanding of how paternal diets affect offspring health in C57Bl6 mice. Different offspring traits were related to different aspects of the paternal diet, suggesting that paternal effects are underpinned by multiple mechanisms and not a simple reflection of under- or overnutrition.

## Results

### Paternal diets affect metabolic and behavioral traits

Paternal metabolic traits were influenced by both additive and interacting effects of all three macronutrients (Fig. 1). The percentage of dietary fat was negatively correlated with weight gain (Fig. 1b, c) and fasting blood glucose (Fig. 2a), while dietary carbohydrate intake was associated with the accumulation of liver triglycerides (Fig. 1d). Differential lipid metabolism is supported by organ weights measured at cull, with mice fed high-fat diets having the lightest gonadal fat deposits (Fig. 2b and Supplementary Fig. 2). Liver, quadriceps and kidney weight were positively correlated with dietary protein (Supplementary Fig. 2). Mice fed low-protein, high-carbohydrate diets had the largest deposits of interscapular brown adipose tissue (Fig. 2c). Insulin sensitivity measured by HOMA-IR was positively associated with low carbohydrate diets (Fig. 1e), particularly when combined with either low or very high protein diets. More specifically, insulin resistance showed a peak at 30% protein, 30% fat, and 40% carbohydrate (corresponding to diet 7; Fig. 1a, e).

Systematically altering dietary macronutrient balance throughout adolescence (treatment diets started at 5 weeks of age)

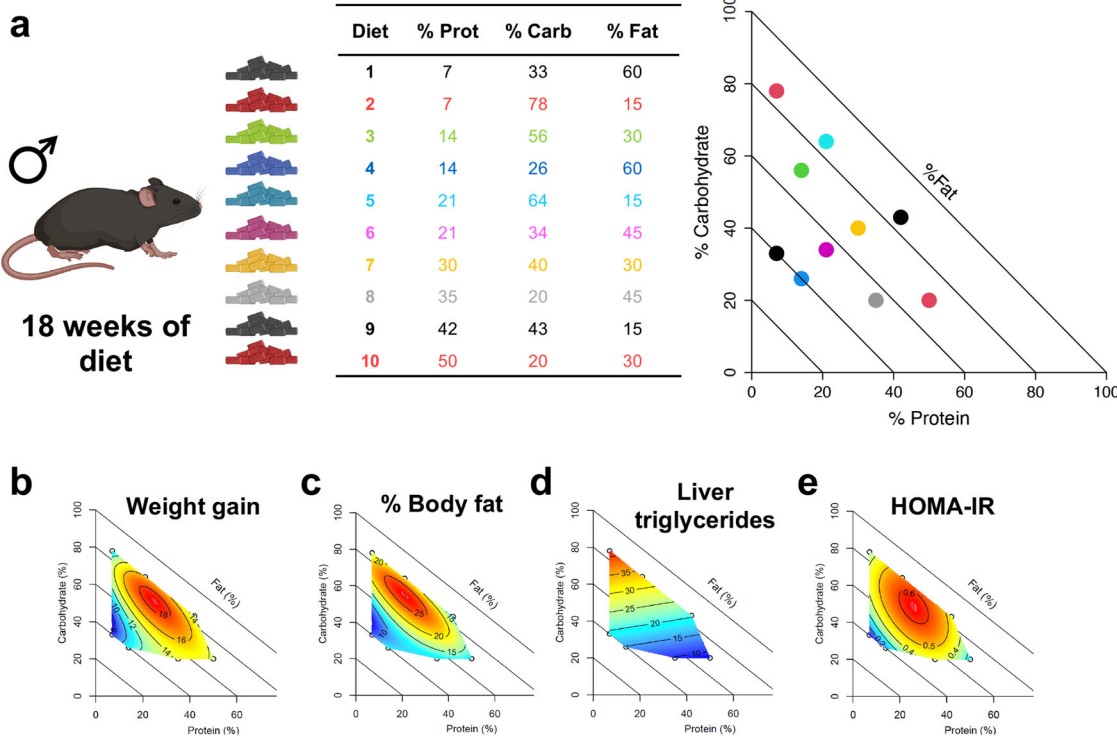

**Fig. 1 | Impact of paternal diets on F0 metabolic phenotypes.** Color scale indicates response trait values (blue = minimum, red = maximum) with isolines showing the model predicted response. Created with BioRender.com. **a** Macronutrient composition (% energy) of paternal treatment diets and map showing location of each diet in the nutrient space plotted as a right angled mixture triangle, **b** increase in body mass (g) after 13 weeks of feeding, **c** Body fat as a percentage of body weight at 18 weeks of age as measured by EchoMRI, **d** Triglycerides measured from snap frozen liver samples (nmoles/mg tissue) and, **e** HOMA-IR calculated from glucose tolerance test. An effect of dietary macronutrient intake on the outcome of interest was inferred,

when the non-null statistical model (i.e., that fitting an effect of macronutrient intake) led to a statistically significant increase in the deviance explained over the null model based on a $\chi2$ test. **a** F0 % Fat Mass $\chi2 = 232.3$, df = 10.04, $p = 4.2\text{e-}8$, $n = 60$, **b** F1 % Fat Mass $\chi2 = 22.29$, df = 14.93, $p = 6.8\text{e-}6$, $n = 60$, **c** F0 HOMA-IR $\chi2 = 1.95$, df = 9, $p = 1.7\text{e-}5$, $n = 60$, (D) F1 % HOMA-IR $\chi2 = 0.4$, df = 9, $p = 0.02$, $n = 60$. Full output from model comparisons is provided in Source Data File 1. Source data for fathers **a**, **c** are provided in Source Data File 1, for female offspring (**b**) in Source Data File 2 and for male offspring (**d**) in Source Data File 3.

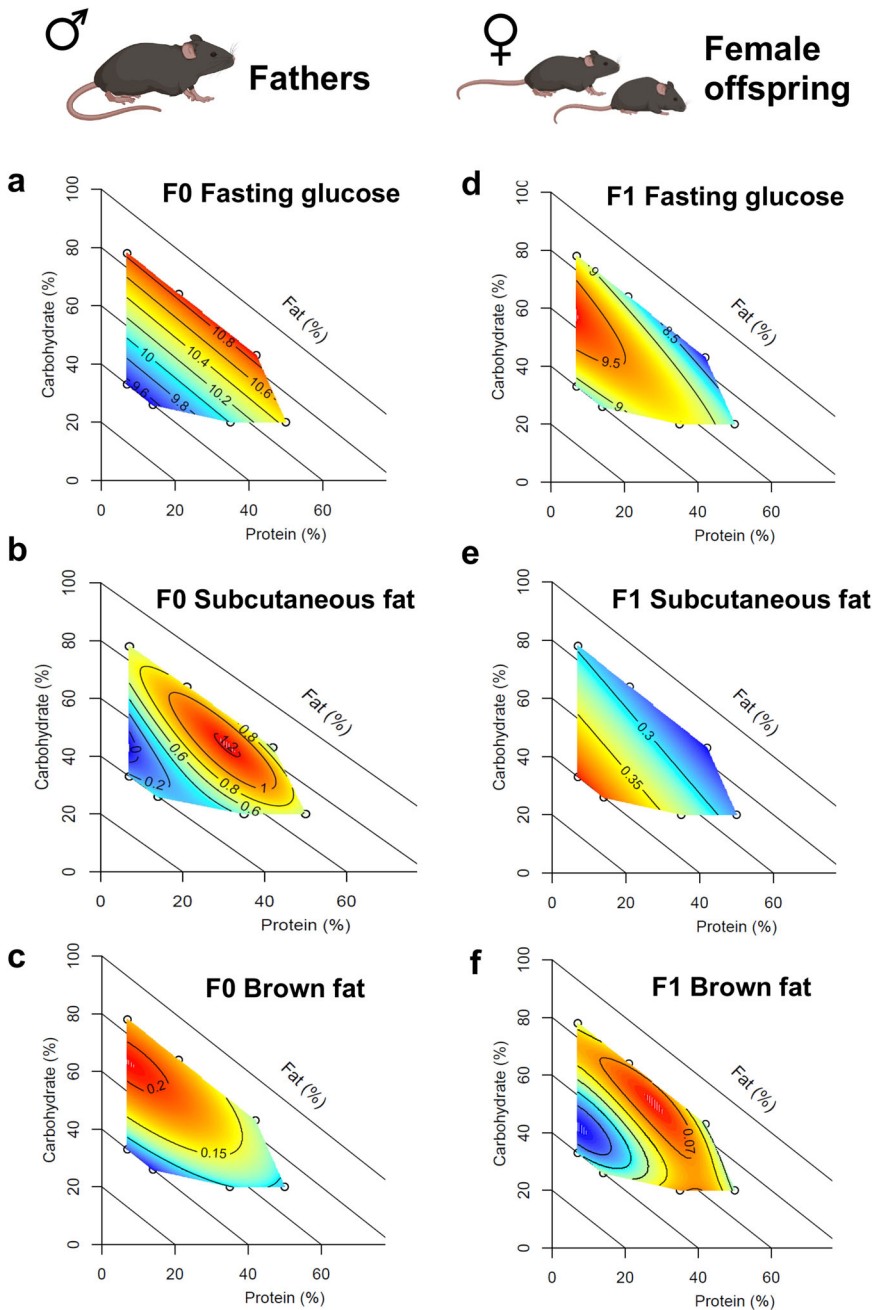

**Fig. 2 | Comparison of metabolic phenotypes in fathers and female offspring.** Note, offspring are plotted against diet composition of fathers. **a, d** Fasting blood glucose (mg/dL) measured during glucose tolerance test at 18 weeks of age, **b, e** weight (g) of subcutaneous white adipose tissue deposits (combined weight of left and right deposit) measured at cull, **c, f** weight (g) of interscapular brown adipose tissue (combined weight of left and right deposit) measured at cull. An effect of dietary macronutrient content on the outcome of interest was inferred, when the non-null statistical model (i.e., that fitting an effect of diet) had an Akaike Information Criterion (AIC) more than two points lower than the null model (i.e., $\triangle$AIC < −2). **a** F0 Fasting Glucose $\triangle$AIC = −2.2, $n$ = 60, **b** F0 Subcutaneous fat $\triangle$AIC = −15.15, $n$ = 60, **c** F0 Brown fat $\triangle$AIC = −16.63, $n$ = 60, **d** F1 Fasting Glucose $\triangle$AIC = −3.51, $n$ = 50, **e** F1 Subcutaneous fat $\triangle$AIC = −6.67, $n$ = 50, **f** F1 Brown fat $\triangle$AIC = −2.51, $n$ = 50. Full model coefficients are given in Supp Table 2. Source data for fathers (**a–c**) are provided in Source Data File 1, and for female offspring (**d–f**) in Source Data File 2. Created with BioRender.com.

also had significant effects on paternal anxiety-like behavior assessed in an elevated plus maze at 15 weeks of age (Fig. 3a–c, Supplementary Fig. 2 and Supplementary Data 2 and 3). Anxiety-like behavior was influenced by the interaction between dietary protein and carbohydrates: mice fed low protein-high carbohydrate diets spent the most time in the closed arms and entered the open arms the fewest times, whereas males fed high protein-low carbohydrate diets showed the opposite behavior pattern (Fig. 3a and Supplementary Fig. 2). The percentage of time spent in open arms was largely determined by dietary protein, with males fed high-protein diets spending the most time in the open (Supplementary Fig. 2). Time spent in the center of the maze, interpreted as the decision-making zone, was positively correlated with dietary fat (Fig. 3b). Dietary effects on the percentage of time spent moving quickly in the maze showed a complex pattern with multiple peaks and troughs in the response surface (Supplementary Fig. 2). Hence, differences in the time spent in each of the arms is not a simple reflection of differences in activity level, supporting the

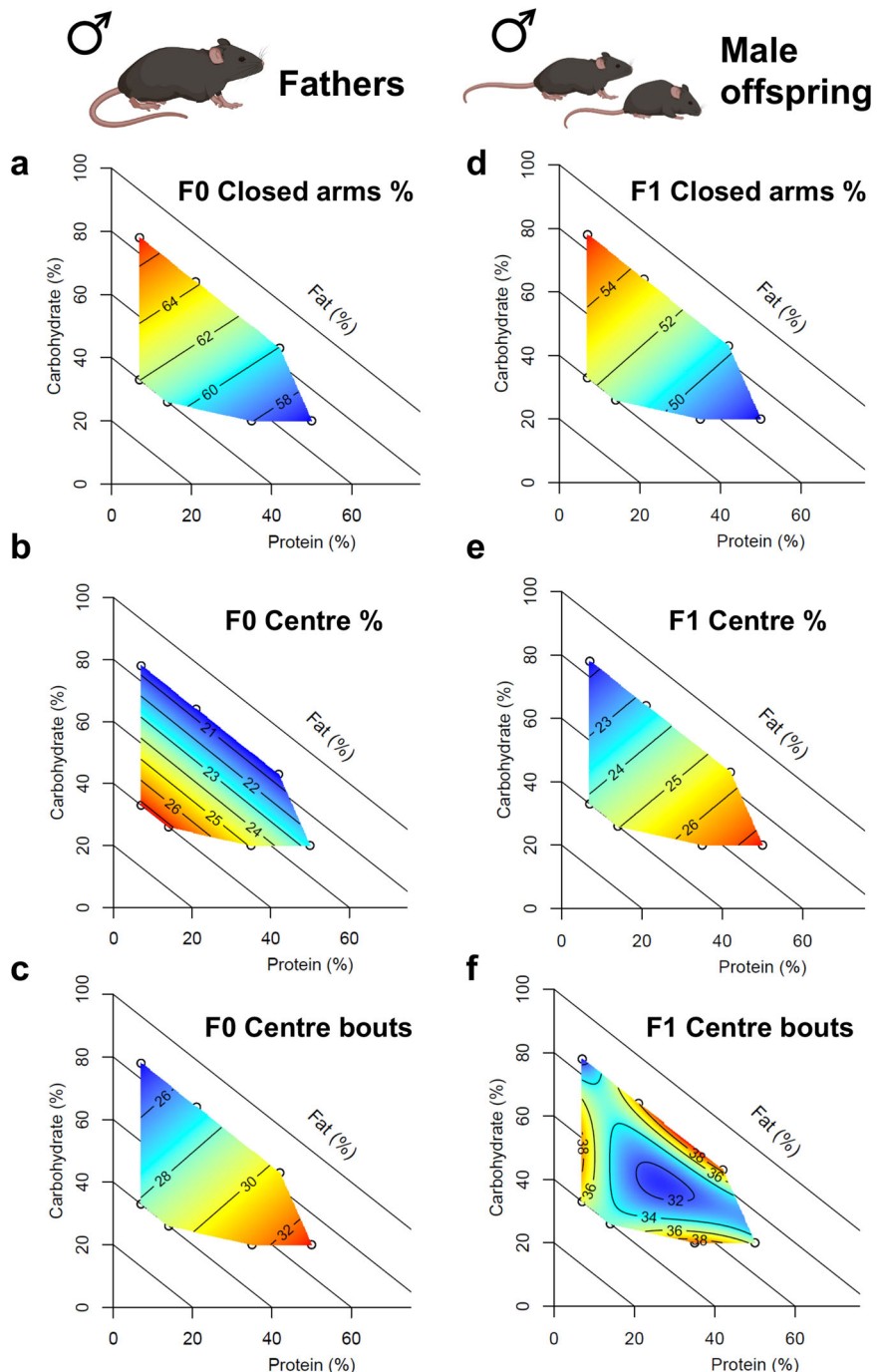

**Fig. 3 | Comparison of behavioral phenotypes in fathers and male offspring.**
Note, that offspring are plotted against diet composition of fathers. Time and activity in different zones of an elevated plus maze, assayed at 15 weeks of age. **a**, **d** percentage of time spent in closed arms, **b**, **e** percentage of time spent in center zone, **c**, **f** number of entries into center zone. An effect of dietary macronutrient content on the outcome of interest was inferred when the non-null statistical model (i.e., that fitting an effect of diet) had an Akaike Information Criterion (AIC) more than two points lower than the null model (i.e., $\triangle$AIC < -2). **a** F0 Closed arm % $\triangle$AIC = -2.19, $n$ = 60, **b** F0 Centre % $\triangle$AIC = -35.33, $n$ = 60, **c** F0 Centre bouts $\triangle$AIC = -9.87, $n$ = 60, **d** F1 Closed arm % $\triangle$AIC = -5.24, $n$ = 50, **e** F1 Centre % $\triangle$AIC = -5.53, $n$ = 50, **f** F1 Centre bouts $\triangle$AIC = -7.33, $n$ = 50. Full model coefficients are given in Supp Table 2. Source data for fathers (**a**–**c**) are provided in Source Data File 1, and that for male offspring (**d**–**f**) in Source Data File 3. Created with BioRender.com.

interpretation that the differences in behavior result from both emotional states of anxiety (time in closed arms) and boldness (time in open arms).

## Paternal diet composition differentially influences offspring metabolic and behavioral traits

Sex-specific effects of paternal diet were detected in both metabolic and behavioral traits in offspring (fed a standard rodent chow diet)

(Table 1). Female offspring showed some changes in metabolic traits, largely driven by differences in paternal dietary fat (Fig. 2). Fasting blood glucose was highest in female offspring of males fed diets containing 21% protein, 34% carbohydrates, and 45% fats (by energy), and lowest when fathers were fed high protein-low fat diets (Fig. 2a). Female offspring of males fed high-fat diets had larger deposits of subcutaneous white adipose tissue, and smaller deposits of inter-scapular brown adipose tissue (Fig. 2b, c). While effects in female

**Table 1 | Summary of model results**

| | F0 | | F1 males | | | F1 females | | | |
|---|---|---|---|---|---|---|---|---|---|
| | MM | GAM | MM | GAM | GAM +litter | MM | GAM | GAM +litter | GAM +Estrus |
| Time in closed arms | 2 | * | 2 | * | * | 1 | - | - | - |
| Closed arm bouts | 1 | - | 5 | - | - | 1 | - | - | - |
| Time in open arms | 2 | * | 1 | - | - | 1 | - | - | - |
| Open arm bouts | 2 | ** | 1 | - | - | 1 | - | - | - |
| Time in centre | 2 | . | 2 | ** | ** | 1 | - | - | - |
| Centre bouts | 2 | * | 5 | * | * | 1 | - | - | - |
| Time moving fast | 4 | *** | 5 | * | * | 1 | - | - | - |
| Body weight | 4 | *** | 1 | - | - | 1 | ** | *** | |
| Body fat | 4 | *** | 1 | - | - | 1 | ** | *** | |
| Lean mass | 3 | *** | 1 | - | * | 1 | * | ** | |
| % body fat | 4 | *** | 1 | - | - | 1 | ** | *** | |
| % lean mass | 4 | *** | 1 | - | - | 1 | - | *** | |
| Fasting glucose | 2 | * | 1 | - | - | 3 | * | ** | |
| Glucose iAUC | 2 | *** | 1 | - | ** | 1 | - | - | |
| Fasting insulin | 4 | *** | 1 | . | . | 1 | . | . | |
| Insulin iAUC | 1 | - | 1 | - | - | 1 | - | - | |
| HOMA | 3 | *** | 1 | * | * | 1 | - | - | |
| Matsuda | 3 | *** | 1 | - | - | 1 | - | - | |
| Liver | 2 | *** | 1 | - | - | 1 | - | - | |
| Liver triglycerides | 2 | ** | 1 | - | - | 1 | ** | * | |
| Gonadal fat | 4 | *** | 1 | - | - | 1 | - | . | |
| Subcutaneous fat | 4 | *** | 1 | - | - | 2 | - | . | |
| Brown fat | 3 | *** | 1 | - | - | 4 | *** | *** | |
| Quad muscles | 2 | * | 1 | - | - | 1 | * | ** | |
| Gastrocnemius | 3 | ** | 1 | - | - | 1 | * | *** | |
| Kidney | 3 | *** | 1 | - | ** | 1 | - | - | |
| Testis | 2 | . | 1 | - | - | | | | |
| Seminal Vesicle | 3 | *** | 1 | - | - | | | | |
| Uterus | | | | | | 1 | - | . | |

Right-side section labels: Elevated Plus Maze (Time in closed arms – Time moving fast); EchoMRI (Body weight – % lean mass); Glucose Tolerance Test (Fasting glucose – Matsuda); Cull weights & tissue analysis (Liver – Uterus).

Mixture model (MM) columns show the MM (Models 1–5) selected by AIC as the best fit to the data. Models increased in complexity from a null model (Model 1) through linear (Model 2), quadratic (Model 3), and cubic (Models 4 and 5) effects of protein, fat, and carbohydrates on the trait of interest. Hence, all models >1 indicate significant effects of treatment diet/paternal diet. Generalized Additive Model (GAM) columns show the significance of treatment diet/paternal diet based on a chi-sq test of a null model versus model including paternal dietary intake (*$p < 0.05$, **$p < 0.01$, ***$p < 0.001$). The significance of paternal dietary intake in GAM models, including litter size and estrus stage on the day of behavior testing, are also shown for offspring traits. Significant effects are highlighted in blue shading.

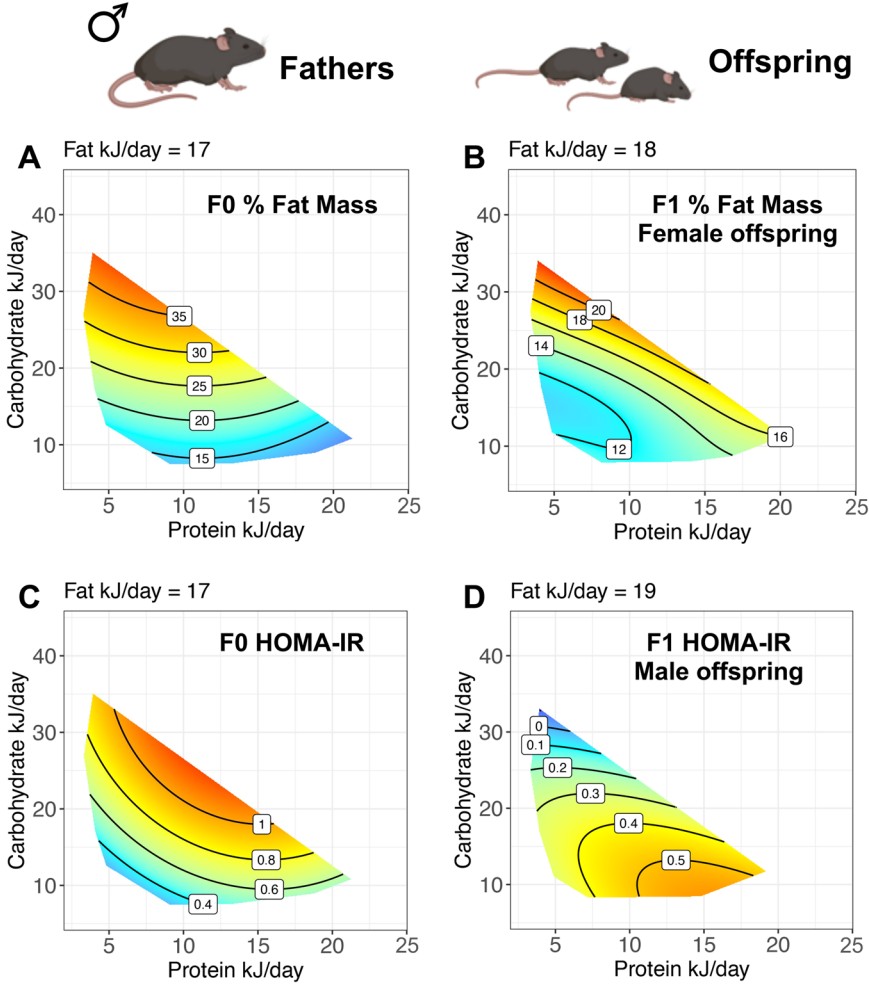

**Fig. 4 | Comparison of paternal and offspring metabolic traits accounting for individual differences in food intake.** GAM predictions of the influence of paternal daily consumption (kJ/day) of Protein and Carbohydrates, sliced through the response surface at the 3rd quartile of Fat intake, on response variables: **a, b** Fat mass as a percentage of body mass measured by EchoMRI at 18 weeks of age, **c, d** HOMA-IR calculated from an oral glucose tolerance test at 18 weeks of age. Note, offspring response surfaces are plotted against paternal intake values. Surfaces sliced through the 1$^{st}$ and 2$^{nd}$ quartile of Fat intake are shown in Fig S4. An effect of dietary macronutrient intake on the outcome of interest was inferred, when the non-null statistical model (i.e., that fitting an effect of macronutrient intake) led to a statistically significant increase in the deviance explained over the null model based on a $\chi 2$ test. **a** F0 % Fat Mass $\chi 2 = 232.3$, df = 10.04, $p = 4.2e{-}8$, $n = 60$, **b** F1 % Fat Mass $\chi 2 = 22.29$, df = 14.93, $p = 6.8e{-}6$, $n = 50$, **c** F0 HOMA-IR $\chi 2 = 1.95$, df = 9, $p = 1.7e{-}5$, $n = 60$, **d** F1 % HOMA-IR $\chi 2 = 0.4$, df = 9, $p = 0.02$, $n = 51$. Full output from model comparisons is provided in Source Data File 1. Source data for fathers (**a, c**) are provided in Source Data File 1, for female offspring (**b**) in Source Data File 2 and for male offspring (**d**) in Source Data File 3. Created with BioRender.com.

offspring were not driven by the same macronutrient compositions compared to effects observed in fathers (compare Fig. 2a–c to 2d–f), male offspring showed changes in anxiety-like behavior that echoed effects observed in the F0, although the variance in traits was greatly reduced (Fig. 3a, d). Male offspring of F0 males fed low protein-high carbohydrate diets spent the greatest amount of time in closed arms (Fig. 3d) and the least amount of time in the center zone (Fig. 3e), with male offspring of high protein-low carbohydrate fed males showing the opposite behavioral pattern. Male offspring of males fed diets with moderate levels of all macronutrients moved the least around the maze, making the fewest entries to the closed arms and center zone (Table 1). Female offspring behavior was not statistically significantly related to paternal diet (Table 1), although it should be noted that females were not cycle-matched at the time of behavioral testing. Accounting for stage of estrus did not qualitatively change the result (Table 1). Time of testing did not account for variance in behavioral traits (e.g. time of day *vs.* time spent in closed arms: F0 $R^2 = 0.022$, male F1 $R^2 = 0.037$, female F1 $R^2 = 0.010$). Outcomes of model selection and raw data are provided in Supplementary Data 2, 4, and 5.

## The incorporation of interacting co-factors unmasks paternal effects on metabolism

Treatment diets were isocaloric by design to disentangle effects of calorie content from macronutrient balance. However, mice were fed ad libitum, and thus calorie intake was not equal across diets due to dietary effects on food intake. In particular, protein leverage effects were apparent, with males fed the lowest percent protein diets consuming the greatest amount of food (Supplementary Fig. 3). Furthermore, although no dietary effects on fertility were observed, paternal diet influenced fecundity, with males fed high-fat diets producing larger litters (Supplementary Fig. 3). Since calories consumed and litter size both have the potential to influence offspring traits, we performed a secondary analysis using a Generalized Additive Model (GAM) that can account for variance in response due to paternal energy intake and litter size effects. Accounting for these moderating factors revealed additional paternal dietary effects, particularly in offspring metabolic traits including insulin resistance and kidney weight of male offspring (Table 1 and Supplementary Fig. 4), and female offspring body composition, liver triglycerides, quadricep and gastrocnemius muscle

weight (Table 1 and Supplementary Fig. 4). As found in previous Western type diet studies, these surfaces reveal that female offspring of male mice that consumed low amounts of protein and high amounts of both fats and carbohydrates had increased body fat (Fig. 4b). Male offspring of male mice that consumed low amount of both fats and carbohydrates had reduced insulin resistance (Fig. 4d). Response surfaces of offspring body composition plotted against paternal dietary intake showed similar trends in male and female offspring, although variance in body weight and body fat in males was less than in females (Supplementary Fig. 4). In contrast, effects of paternal dietary intake showed distinctly different patterns in male and female offspring glucose tolerance, with variance in metabolic traits among male offspring greater than female offspring (Supplementary Fig. 3). Results of model comparisons are provided in Supplementary Data 6.

## Discussion

Here, we report that paternal dietary macronutrient balance influences offspring metabolic and behavioral traits, largely in a sex-specific fashion. All three macronutrients were found to have additive and interacting effects on a range of paternal and offspring traits. Differences among chow-fed offspring in metabolic traits (including body composition) were largely driven by the proportion of fat in the paternal diet, with stronger effects generally observed in female offspring. In contrast, behavioral traits were largely influenced by the interaction between paternal dietary protein and carbohydrates and were only observed in male offspring. Incorporating individual differences in paternal food intake (and thus differences in calories consumed) into model predictions unmasked the combined effect of macronutrient balance, food intake and offspring litter size on metabolic traits in both male and female offspring.

When offspring traits were mapped against the proportion of each macronutrient in the paternal diet, female offspring adiposity was largely explained by the percentage of fat in the paternal diet (Fig. 2). Female offspring of males fed high-fat diets had larger subcutaneous fat deposits and smaller brown fat deposits. Similarly, Fullston, et al.[31] and Huypens, et al.[32] found paternal high-fat diets induced an increase in adiposity of female but not male C57BL6 adult mouse offspring. While effects of paternal dietary fat on offspring adiposity were largely linear, fasting glucose showed an intermediate peak, with female offspring of males fed both high-fat and low-fat diets showing reduced fasting glucose levels. An effect of paternal dietary protein was also evident, with reduced blood glucose observed in female offspring of high-protein diet fathers. These observed differences in female offspring metabolic traits did not reflect patterns observed in fathers, where the proportion of dietary fat was generally found to be negatively correlated with body fat and insulin sensitivity (Fig. 1). These results match studies of under-nutrition, where lean fathers have produced fatter offspring[33–35].

If looked at superficially, our results appear to contradict those from low-protein diet paternal studies, where effects on offspring metabolic traits are found despite dietary fat content remaining constant[33,35]. However, these differences are reconciled when we integrated paternal food intake into the models. Because the proportion of dietary protein affects the amount of food consumed in an unconstrained food environment[26,28,29], plotting predicted offspring responses against paternal macronutrient intake reveals that adiposity of female offspring is highest when fathers have consumed low amounts of protein and high amounts of carbohydrates, even when the response surface is sliced at a single plane of fat intake (Fig. 4). This demonstrates how apparent differences among studies can be resolved by increasing the resolution of our models to incorporate interacting factors.

In contrast to metabolic traits, behavioral traits were solely determined by interactions between paternal dietary macronutrients. Male offspring of males fed low protein-high carbohydrate diets spent more time in the safety of closed arms of the elevated plus maze, with no influence of the percentage of paternal dietary fats predicted from the favored model (isolines run perpendicular to fat axis in Fig. 3d). This pattern of time spent in different zones of the elevated plus maze reflected direct effects of diet on fathers, although the range of effect is greatly reduced in offspring (Fig. 3). Bodden, et al.[36] similarly found no difference in paternal or offspring anxiety-like behavior of C57BL6 mice fed control versus 'Western' high-fat/high-sugar diets. However, Korgan, et al.[37] found in Long-Evans hooded rats, both fathers fed high-fat diets and their offspring spent less time in the open arms of an elevated plus maze. Importantly, in Korgan, et al.[37], rats were held under reverse light cycle conditions, and behavioral testing was conducted under red light during the active phase. Although diurnal phase does not appear to alter anxiety-like behavior as measured in rodent behavior assays[38,39], it is not known whether circadian rhythms interact with nutritional factors to influence behavior.

Another nutritional factor that should be considered when interpreting results of paternal effects studies is macronutrient quality[40]. For example, the relative balance of amino acids within the protein component of diets can influence appetite and metabolic health[41], and likewise the types of carbohydrates in the diet[42]. In our study, amino acid balance was optimized by matching proportions to the 'exome'[43], which is expected to reduce the reproductive impacts of low-protein diets. In addition, the quality of dietary fats in our study was kept consistent across diets by maintaining the proportion of saturated fatty acids at 23.2% of total lipid content and the relative amount of omega-3 to omega-6 fatty acids at a ratio of 1: 3.7. Western diets typically contain excessive amounts of saturated and trans-fatty acids and a deficiency of omega-3 fatty acids[44]. Supplementation of omega-3 fatty acids in the paternal diet can affect offspring brain function[8], and thus differences in the balance of dietary fatty acids among studies may also contribute to differences in offspring behavioral outcomes. Finally, while vitamin content was kept consistent across diets in our study, paternal diet vitamin supplementation can influence offspring outcomes in context-specific ways[34]. Hence, while our study was designed to investigate paternal dietary effects of macronutrient balance in high-quality diets, future studies should investigate how these effects differ in the context of unbalanced nutrient mixtures.

We acknowledge that while it would be ideal to examine the context-dependence of effects observed in this study, logistical considerations limit our ability to test multiple factors simultaneously. Due to the number of diets and thus number of animals involved in Nutritional Geometry Framework experimental designs, it was not logistically feasible to test offspring in different food environments or at multiple timepoints. We also did not have the power to test for interactive sex effects in our study, and therefore male and female offspring were examined separately. Hence, this study should be considered as a first step, designed to illustrate how moving beyond two-diet experimental designs can uncover novel insights and inspire future research directions. Based on results from the present study, follow-up experiments can be designed to examine specific nutrient effects using fewer diet comparisons, allowing for context-dependence of results to be tested, increasing precision and utility of dietary recommendations. For instance, on the basis of our results, we would recommend that isocaloric diets that vary quantity of fat, but fix the ratio of protein to carbohydrate, make a good model for understanding the effects of paternal diet on metabolism in female offspring.

The influence of the paternal environment prior to conception on offspring traits is often overlooked. While research into environmental paternal effects has increased in recent years, studies have focused on elucidating the mechanisms of epigenetic inheritance, and not in the identification at the critical environmental determinants of paternal effects. Models of paternal effects are

largely designed with a standard environmental versus an extreme environment of stress of various nutritional or psychological nature. Thus, our perspective on the consequences of environmental variance remains narrow. We demonstrated that the proportion of fats in the paternal diet explained differences in offspring metabolic traits, but effects of paternal dietary protein to carbohydrate ratio were also revealed once differences in food intake (and therefore calories consumed) were integrated into models. By examining additive and interactive effects of paternal dietary macronutrients across a broad (yet physiologically relevant) nutrient space, and by considering effects through the lens of both dietary composition and nutrient intake, we were able to demonstrate how models of both over- and under-nutrition using different dietary paradigms can induce similar phenotypic responses in offspring.

In conclusion, our study uncovered a dynamic interplay between preconceptional dietary composition and energy intake in paternal inheritance. Using the Nutritional Geometry Framework, we identified relationships between the consumption of certain macronutrients and offspring phenotypes in mice. Our mapping of offspring responses to complex paternal diets provides the foundations for future research directions in the development of preconceptional dietary guidelines for males.

## Methods

### Animals and ethical approval
All procedures were reviewed and approved by the University of Sydney Animal Ethics Committee (project number 2019/1610). C57BL/6 J mice (JAX strain code 000664), purchased from Animal Resources Centre (Murdoch, Australia), were used in this study. Mice were housed, and experiments undertaken in a dedicated pathogen-free facility (24–26 °C, 44–46% humidity, 12 h day/light cycle) in the Charles Perkins Centre (Sydney, Australia). A subset of males studied here were used in previous investigations (Crean et al.[45] Cohort 1; Farris et al., in review). Reproductive traits of paternal mice included in this study are reported in ref. 45, Cohort1. Animal health was checked at least twice weekly, and body weights measured weekly.

### Experimental diets (Fig. 1a)
Diets were co-designed with and manufactured by Specialty Feeds (Glen Forrest, Australia). All diets were based on ingredients of the semi-pure AIN93G standard rodent chow ( ~ 19% protein, ~18% fat, ~63% carbohydrate)[46], but the proportion of ingredients was altered to systematically manipulate the macronutrient balance across the range of physiologically viable nutrient intakes (Fig. 1a, Supplementary Data 1). Diets were isocaloric (3.5 kcal/g), achieved by adjusting the amount of non-digestible cellulose. Protein content was exome-matched to the mus musculus genome[43], achieved by mixing casein and whey protein isolates supplemented with leucine, threonine, methionine, tyrosine, phenylalanine, tryptophan, alanine, aspartic acid, arginine, glycine, histidine and serine. Omega 3 to omega 6 fatty acid ratio was maintained at 1:3.7 using a combination of soybean oil, linseed oil and lard, with saturated fats making up 23.2% of dietary fats. Carbohydrate sources included wheat starch, dextrinised starch and sucrose at a ratio of 4:1.3:1. Micronutrient content was kept consistent between diets.

### Experimental timeline (Supplementary Fig. 1)
Sixty 4-week-old male mice were acquired, housed in groups of 3 animals per cage, and allowed to acclimate for 4 days before being switched from standard chow (Specialty Feeds standard diet SF00-100, 14.0 MJ/kg, 23% protein, 11% fat) to experimental diets fed ad libitum ($n = 6$/diet). After 10 weeks on diet, anxiety-like behavior was assayed in an Elevated Plus Maze (details below), and then males were separated into individual cages in preparation for mating. After 12 weeks on diet, males were mated to age-matched,

fertility-proven (previously produced a litter) C57BL/6 J females (details below). Unsuccessful pairs were given a second chance at mating 2 weeks later ($n = 13$), and a final opportunity 2 weeks later if still unsuccessful ($n = 10$). Body composition and glucose tolerance was assayed at 18 weeks of age (details below). Males were culled after the successful birth of pups or after the third mating attempt. Large litters were culled to 8 pups at 3 days old to minimize litter size effects, and pups were weaned on to standard chow (SF00-100) at P21. Behavior and metabolic assays were performed on offspring at the same age as they were performed on fathers for comparative purposes. Two males and 2 females from each litter were assayed where possible ($n = 99$ male and 94 female offspring), with the average trait values per sex per litter used in analyzes. Litter mates were kept as companion animals to maintain group housing requirements as per ethics permit.

### Mating
Females were individually housed, fed standard chow (SF00-100), and exposed to bedding from their partner's cage prior to mating. Males were given the opportunity to mate overnight for a maximum of four consecutive nights each mating cycle and returned to home cages each day to minimize exposure to standard chow. Females were checked for copulation plugs each morning. Once a copulation plug was observed, females were left undisturbed until a litter was born, or a subsequent round of mating was initiated. Eight pairs failed to produce any offspring from the three mating cycles (diet 5 $n = 3$, diet 6 $n = 1$, diet 9 $n = 3$, diet 10 $n = 1$).

### Metabolic phenotyping
Body composition was measured at 18 weeks of age (13 weeks after commencement of treatment diets in the F0) by quantitative magnetic resonance using an EchoMRI-900-A130 (EchoMRI, Houston, USA). Oral glucose tolerance tests were performed in the same week. Mice were fasted from 9am on the day of testing, and the procedure commenced at 1 pm. Blood glucose was measured using a clinical glucometer (Accu-Chek Performa, Roche Diagnostics Australia Pty Ltd) from blood droplets obtained by tail tipping performed 15 mins prior to the administration of glucose (single wound, no serial tipping required). Glucose (2 g/kg lean mass calculated individually per mouse) was administered via oral gavage. Blood glucose was measured at baseline, 15, 30, 45, 60, and 90 min, and the incremental area under the curve (iAUC) was calculated. Blood from tail tipping was also used to measure blood insulin at baseline, 15 and 30 min, using an enzyme-linked immunosorbent assay (ELISA) following the manufacturer's instructions (Crystal Chem IL). Food intake of fathers was measured by weighing food in and out of individual cages over two 24 h periods at 16 and 20 weeks of age (11 and 15 weeks after commencement of treatment diets). Bedding was changed at the start of intake measures and sifted for food crumbs to obtain as accurate measures of food consumed as possible. The average of both measures was used in analyzes.

### Behavioral phenotyping
The behavior of mice was assayed in an elevated plus maze at 15 weeks of age (10 weeks after commencement of treatment diets in the F0). The maze was elevated 50 cm above the ground and consisted of 2 closed arms (30 × 5 cm), 2 open arms (30 × 5 cm), and a central zone (5 × 5 cm). The light intensity on the open arms of the maze was 700–750 lux, and in the closed arms was 175–225 lux. Each mouse was placed into the center of the maze facing the left open arm for consistency at the beginning of recording using Logitech webcam software (Logitech; Lausanne, Switzerland). Mice were exposed to the elevated plus maze for a total of 5 min and the apparatus was cleaned thoroughly with ethanol between animals. TopScan Image Analysis Software (CleverSys Inc; VA, USA) was used to calculate the amount of time spent in the

center, open and closed arms of the maze, and entries into each zone. An entry was defined as the mouse crossing the dividing line between zones with all four feet. Activity level was assessed by the percentage of time spent moving > 20 mm/s ('fast' motion).

## Tissue collection

At sacrifice, animals were anaesthetized with sodium pentobarbital (100 mg/kg) and exsanguination by cardiac puncture was performed. Livers were dissected, weighed, snap frozen in liquid nitrogen, and stored at −80 °C for later analysis. Gonadal and subcutaneous white adipose tissue, interscapular brown adipose tissue, quadricep and gastrocnemius muscles, kidneys, testes, seminal vesicles, and uterus were removed and weighed. The average of both kidneys, testes and seminal vesicles were used in analyzes.

## Triglyceride Assay

Frozen liver tissue (20–30 mg) was cut (weight recorded) and transferred into chilled 2 mL microcentrifuge tubes containing 5 mm stainless-steel beads. Two rounds of homogenization in 1 mL of 2:1 Chloroform: Methanol (C:M) solution were performed using a Qiagen TissueLyser LT (50 Hz x 1 min). Homogenates were transferred to 7 ml tubes, and a further 3 mL of C:M solution added making a total volume of 4 mL. These samples were left to extract on rotating mixers overnight at 4 °C. The following day, 2 mL of 0.6% NaCl solution was added and samples briefly vortexed before being centrifuged for 10 min at 2000 rpm for phase separation. The lower lipid-containing phase of the resulting biphasic system was transferred to test tubes using glass pipettes and dried using a nitrogen gas apparatus. The dried lipids were then dissolved in 500 µl of pure ethyl alcohol (Sigma Aldrich: E7023-500ML) and vortexed to resuspend. Samples (5 µL) were plated in duplicate for quantification using a colorimetric assay in a 96-well plate using known concentrations (0–22.9 nmoles/well) of Precimat Glycerol (2.29 mmol/L) as standards (Roche: 10166588). After drying for 20 min at 37 °C, samples were incubated with 300 µl of Triglyceride Infinity (Thermo Scientific: TR22421) per well at 37 °C for 30 min. Absorbance was measured using a TECAN Infinite M200 Pro Plate Reader at 490 nm.

## Statistical analysis

Data were analyzed in R (v 4.2.0) using mixture models[47] (mixexp package) to account for the proportional nature of the dietary macronutrient balance in the experimental design. To test for interactive and non-linear effects a total of 5 models, including a null model, were fitted for each experimental variable and tested for linear, quadratic and cubic effects of protein, fat and carbohydrate. Model selection was based on the lowest relative Akaike Information Criterion (AIC) as a measure of goodness of fit. For data visualization, the selected predicted model was plotted as a right-angled mixture triangle response surface[30] (Fig. 1). Offspring traits were plotted against the paternal diet. AIC values and a summary of the selected model is provided for all variables in Table S2 and raw data for all outcome variables is given in Tables S3–5. As a secondary analysis, response variables were analyzed using GAM models (which account for individual differences in food intake), with predictions from models visualized as response surfaces[48]. Chi-sq tests were used to compare between null and intake models, either including or excluding litter size as a co-variate for offspring traits. Results of model comparisons are provided in Supplementary Data 6.

## Reporting summary

Further information on research design is available in the Nature Portfolio Reporting Summary linked to this article.

## Data availability

Source data are provided in this paper. All of the information required to reproduce the results included here can be found in the text, figures, and supplementary information. Source data are provided in this paper.

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

## Acknowledgements

Victoria Pye is acknowledged for her animal husbandry and technical assistance. We thank the members of the Nutritional Biology group at the University of Sydney and members of the GECKO consortium for discussions that helped form ideas and interpretations of the results of this study. This work was supported by a Challenge Programme Grant from the Novo Nordisk Foundation (NNF18OC0033754) to the Gametic Epigenetics Consortium against Obesity (GECKO). The Novo Nordisk Foundation Center for Basic Metabolic Research is an independent research center at the University of Copenhagen, partially funded by an unrestricted donation from the Novo Nordisk Foundation (NNF18CC0034900). This work was supported by the French Government (National Research Agency, ANR) through the "Investments for the Future" programs LABEX SIGNALIFE ANR-11-LABX-0028-01 and IDEX UCAJedi ANR-15-IDEX-01.

## Author contributions

A.C., M.N., R.B. and S.S. designed the study. A.C., T.F., T.C., F.M., G.A. and T.P. performed the experiments. A.C., A.S., M.N., R.B. and S.S. analyzed the results. A.C., A.S., R.B. and S.S. wrote the manuscript. All authors approved the final version of the manuscript.

## Competing interests

The authors declare no competing interests.
