## [Peer Review File · Nature Communications]

Paternal dietary macronutrient balance and energy intake drive metabolic and behavioral differences among offspringREVIEWER COMMENTS

Reviewer #1 (Remarks to the Author):

Using a murine model, the submitted paper by Crean and colleagues has carefully assessed the role of specific paternal diets on his own health as well as that of male and female offspring.

Since the father is often overlooked with regard to pregnancy and offspring health, the paper is a welcome addition to the field. The methods appear sound and the interpretation of the data is appropriate. However, this reviewer suggests the following:

1. Reorganization. The paper was somewhat difficult to follow and may be better received if the paternal data is presented separately from the offspring data. Specifically, it would be best to combine the F0 data currently shown in Figure 1 with his part of Figures 2 and 4 (perhaps one large figure or two smaller figures). Next, the female offspring data currently shown in Figure 2 could be presented, followed by the male offspring data in Figure 3. Data in Figure 4 makes sense as is and should follow the offspring data. Finally, Table 1 should be presented last since it is a summary.
2. The legend for the Table needs to include the definition of GAM. It might also be helpful within the legend to initially define the Mixture Models as (MM, Models 1-4) to be clearer. Keep the model descriptions which are presented on the next line. Additionally, please replace "litter effects" with "litter size" as the term "litter effects" typically refers to the phenomena that littermates tend to be more similar than non-littermates.
3. This reviewer takes issue with the sentence on lines 249-250 regarding how recently the paternal environment has been shown to impact offspring health. Although, as stated above, fathers are not as well studied as mothers in this regard, the role of the father in placental development has been known for nearly forty years (PMID: 3834032) and numerous studies have demonstrated the impact of paternal factors (age, obesity, environmental exposures) on placental health and offspring health (eg, adiposity and behavior traits). Please revise this sentence.
4. It is interesting that the majority of mating pairs that failed to achieve pregnancy were maintained on the diets with the lowest amount of fat (Diets 5 and 9, both with 15% fat). It is standard practice in mouse husbandry to provide a high fat diet to the female, but

presented data suggests this may be important for the male as well.

5. The concluding statement should be modified. There has been robust research in humans regarding diet and fertility (eg, PMID: 28844822). While the current data adds to the field with regard to the importance of diet in contributing to paternal health and offspring effects, the specific dietary requirements of a human population are likely to be different from those of mice.

Reviewer #2 (Remarks to the Author):

The manuscript provides an organized assessment of dietary macronutrient composition influences on paternal and offspring outcomes. This appears to be an expansion of previous work in the nutritional geometry framework space that incorporates consideration of the inherent correlation between macronutrients within two-diet comparisons studies due to macronutrient balancing effects. The study design provides the opportunity for unique insight related to nutrition research.

General Comments:

Study design and interpretation broad considerations

- 10 isocaloric diets tested, but the amount of intake was not matched for paternal among groups.
- While two 24-hr measures of food (energy) intake were collected and utilized in some of the analyses, with outcome significance altered when incorporating the measures, it is not clear whether such short-term measures can reliably reflect individual overall energy intake.
- It is not clear whether food intake for offspring monitored or included in assessments?
- Mothers were kept on 'standard rodent chow' – definition of diets – chow vs. semi-purified should be aligned with the broader animal nutrition literature.
- Diet of maternal is understandably fixed, but may limit the ability to interpret alignment between paternal and maternal-specific effects and generalizability of findings to other scenarios (e.g. maternal-paternal dietary congruence, what if both father and mother had a high-fat, low-carb diet)?

- One housing temperature utilized, as it is known that temperature affects energy intake, macronutrient selection and reproductive outcomes, there exist challenges for broad interpretation of findings.
- Extrapolation of findings for clinical relevance to dietary guidelines premature and incomplete, overreaching.

Specific Comments

ARRIVE guidelines throughout (e.g. should species be listed in the title, details in figure legends regarding species, sample sizes, etc.?)

Nutritional Geometry framework – capitalization and consistency?

line 35 – Definition of ‘environmental variability’?

Lines 43-46, not sure this follows...all of the above could share something in common beyond the 'stress' interventions.

Line 62-64 - Not clear that this is possible within the context of research study utilized here for rodents...

Line 94 – clarify the ‘adolescence’ stage within the model used for general audience – the ages of animals are noted in the methods and overview figure, but the developmental stage by age of mice may not be known to all readers?

Line 98-99 and elsewhere – nomenclature/wording... (definitions of ‘high-protein’ ‘low-carbohydrate’, etc. can differ among studies, and although the overall diets are described for macronutrient proportion contents the interpretation of ‘high’ or ‘low’ may not always be self-evident to individual readers).

Line 111 and following (and Line 345 -357) – the collected and presented data regarding ‘behavioral’ outcomes is limited to a single assay and timepoint of measure. The phraseology of ‘metabolic and behavioral traits’ accurately reflects the combined

assessment of metabolic outcomes with the single behavioral assessment, but possibly unintentionally results in the reader expecting more than one behavioral assessment to be presented. As other behavioral assays are available and the elevated plus maze reliance on 'activity', the presented data offer a specific/narrow assessment of a behavioral trait of the animals. The time of day of measures should also be noted given the sample size utilized and duration of assay performance (it is noted the authors reference publications regarding diurnal phase impact on behavior assays, it is not clear that time of assessment on behavior assay outcomes should be disregarded in analyses to be the consensus in the field).

Line 169-172 – using individual differences in paternal food intake /calorie consumed – while this was measured, is there any existing knowledge regarding how acute feeding differences can influence epigenetic effects on sperm – e.g. does 1 meal have acute effects or does feeding need to persist for a full day, 1 week, 1 month, etc.? Similarly, if you do not know the amount of intake from the 'brown chow' when paired with females for mating, could this have acute effects on sperm? Would an overview of murine spermatogenic cycle help place the results in context and/or are there windows within the spermatogenic cycle which are more important than others?

Line 242 – while the presented results are potentially intriguing, it may be a bit presumptuous to consider the work foundational versus more preliminary in light of the overall design and sample sizes, with inherent limitations for systematically modifying multiple factors which would be expected to interact with dietary effects.

Lines 244-247 - what would the authors propose are the fewer diet comparisons that would be optimal - any attempt to identify the optimal ratio/balances to cover the spectrum sufficiently to address the context-dependence in future studies?

Line 268-269 – 'foundations of preconceptional dietary guidelines for males' – Overreaching. Should be extensively revised or removed?

Line 305- what is the specific definition of 'proven ... females'? Were there any differences in pre-study reproductive outcomes for determination of reproductive proven status for the

females utilized despite being 'age-matched' to males?

Line 308 - age and exposure differences then – are any of these incorporated in the analyses?

Line 309 - was litter size adjusted before birth or only after? How separate the programming effects in vivo during pregnancy/development from litter size, etc. versus the paternal diet effects?

Line 317-320 – breeding scheme - appears there was food in the cage at night during the mating trial, but not quantified regarding intake and rather stated 'returned to home cages each day to minimize exposure to brown chow', yet most intake occurs during the night – if not quantified, how does one then know or not whether any effect?

Line 323 - half of the sample for two of the diets failed to produce any offspring from 3 mating cycles? Similarly, was the consideration of 1, 2 or 3 mating cycles or failure to reach reproductive success considered in the analyses for outcomes?

Line 329 – based on the light:dark cycle reported, food removal from 9a-1p would likely be even longer period of food deprivation (with 6a:6p light cycle), light phase then during normal sleep/non-feeding state?

Line 338-340 – for the two 24-hour periods of food intake measurement at 16 and 20 weeks of age, does this mean one 24-hr measure within week 16 and one 24-hr measure within week 20 of age or two 24-hr measures performed each of those two weeks (e.g. two 24-hr measures in week 16 and two 24-hr measures in week 20 of age)? Food intake can be quite variable from day-to-day and is sensitive to different types of environmental disturbances (e.g. cage change, outcome measurement interaction, etc.). With the 'average of both measures' being used in the analyses (line 342) is there any knowledge of how representative the value was of overall intake behavior?

Lines 386-399 – It appears the diet composition assessments consider the macronutrient

composition. It also appears dietary fiber was differential across some dietary macronutrient balances as it appears to be utilized to help maintain dietary energy density. Considering the difference in fiber among groups, was fiber also considered in the analyses of outcomes?

Line 428 – Table 1 – how was multiple testing addressed? Red may also not be an optimal shading color given its role in fold and direction changes in other uses (e.g. microarray, etc).

Table on 430 – what does column 4 (first column under F1 males) represent via the number - what is 5? Does 1-4 represent the models?

Line 443 – Fig 2 - why metabolic for female offspring and not male offspring as well? b/e weight of 'paired' subcutaneous, c/f weigh of paired interscapular – meaning of 'paired'? How distinguish brown adipose tissue activity from weight of dissected tissue?

Line 440 - Fig 3 - Why one sex assessed for offspring – rationale for choice?

Line 457 – Fig. 4 - why chose the 3rd quartile – how determination made?

Suppl Figs

Sup Fig 1 – expand legend and clarify abbreviations used, how adjust for number of attempted matings?

Sup Fig 2 – clarify in the figures which animals these represent and time of measure?

Sup Fig 3 – Some diets have 50% higher intake among paternal cohort?

Sup Fig 4 – Useful information that could use clarification and context in figure legend?

REVIEWER COMMENTS

Reviewer #1 (Remarks to the Author):

Using a murine model, the submitted paper by Crean and colleagues has carefully assessed the role of specific paternal diets on his own health as well as that of male and female offspring. Since the father is often overlooked with regard to pregnancy and offspring health, the paper is a welcome addition to the field. The methods appear sound and the interpretation of the data is appropriate.

However, this reviewer suggests the following:

1. Reorganization. The paper was somewhat difficult to follow and may be better received if the paternal data is presented separately from the offspring data. Specifically, it would be best to combine the F0 data currently shown in Figure 1 with his part of Figures 2 and 4 (perhaps one large figure or two smaller figures). Next, the female offspring data currently shown in Figure 2 could be presented, followed by the male offspring data in Figure 3. Data in Figure 4 makes sense as is and should follow the offspring data. Finally, Table 1 should be presented last since it is a summary.

We appreciate that there are several ways to cluster the results, including by generation as suggested by the reviewer. We have seriously considered to present our results in a sequential generation-specific manner, but we felt that it was most important to display the paternal surfaces alongside the matching offspring surfaces, as it allows for easy visual comparison of phenotypic traits in F0 versus F1. We think this allows the reader to see when F1 traits reflect paternal traits (for example Fig 3a/d and Fig 4a/b), and when phenotypic patterns in the F1 act in opposite directions to paternal traits (for example Fig 2b/e and Fig 4c/d). The alternative approach suggested would require the reader to scan back and compare plots across different figures (and sometimes different physical sizes) making contrasting F0 and F1 very challenging.

We agree that Table 1 should be presented last and will note this for the production editors.

2. The legend for the Table needs to include the definition of GAM. It might also be helpful with the legend to initially define the Mixture Models as (MM, Models 1-4) to be clearer. Keep the model descriptions which are presented on the next line. Additionally, please replace “litter effects” with “litter size” as the term “litter effects” typically refers to the phenomena that littermates tend to be more similar than non-littermates.

We thank the reviewer for this suggestion. We have amended the table legend.

3. This reviewer takes issue with the sentence on lines 249-250 regarding how recently the paternal environment has been shown to impact offspring health. Although, as stated above, fathers are not as well studied as mothers in this regard, the role of the father in placental development has been known for nearly forty years (PMID: 3834032) and numerous studies have demonstrated the impact of paternal factors (age, obesity, environmental exposures) on placental health and offspring health (eg, adiposity and behavior traits). Please revise this sentence.

The reviewer is right, the discovery of paternal effects is not recent. We meant that until recently, the field of developmental biology was assuming that preconceptional factors in fathers had minimal effects on the offspring. We have clarified what we meant in the revised text lines 247-252.

4. It is interesting that the majority of mating pairs that failed to achieve pregnancy were maintained on the diets with the lowest amount of fat (Diets 5 and 9, both with 15% fat). It is standard practice in mouse husbandry to provide a high fat diet to the female, but presented data suggests this may be important for the male as well.

The reviewer has rightly noted that intriguing observation. It would be indeed interesting to test if this phenomenon is robust using a design of appropriate power, which was not the case in the present study. For this reason, we do not discuss this finding in the current manuscript.

5. The concluding statement should be modified. There has been robust research in humans regarding diet and fertility (eg, PMID: 28844822). While the current data adds to the field with regard to the importance of diet in contributing to paternal health and offspring effects, the specific dietary requirements of a human population are likely to be different from those of mice.

We have revised our concluding statement to clarify that our study was conclusive in mice but only formed the foundation for the development of dietary guidelines for humans. We now state: “[...] we identified relationships between the consumption of certain macronutrients and offspring phenotypes in mice. Our mapping of offspring responses to complex paternal diets provides the foundations for future research directions in the development of preconceptional dietary guidelines for males” (lines 265-269).

Reviewer #2 (Remarks to the Author):

The manuscript provides an organized assessment of dietary macronutrient composition influences on paternal and offspring outcomes. This appears to be an expansion of previous work in the nutritional geometry framework space that incorporates consideration of the inherent correlation between macronutrients within two-diet comparisons studies due to macronutrient balancing effects. The study design provides the opportunity for unique insight related to nutrition research.

General Comments:

Study design and interpretation broad considerations

- 10 isocaloric diets tested, but the amount of intake was not matched for paternal among groups.

The reviewer is right, the amount of intake was not matched across groups. We provided mice with unrestricted access to food, and measured food intake of F0 mice. As macronutrient composition has an intrinsic effect on food intake, matching amounts of intake could potentially introduce confounding factors. For example, matching intake across groups may constitute fasting for certain groups. Our experimental approach allowed us to uncover interactive effects of dietary composition on both food intake and paternal traits, revealing novel insights into effects of paternal diet composition versus energy intake on offspring health.

- While two 24-hr measures of food (energy) intake were collected and utilized in some of the analyses, with outcome significance altered when incorporating the measures, it is not clear whether such short-term measures can reliably reflect individual overall energy intake.

Thank you for encouraging us to think about this issue. The two-individual 24h food intake estimates were significantly correlated ($p < 0.001$), although individual variability between the estimates was observed (as expected) ($R^2 = 0.407$). The relationship between the two food intake measures is now shown in Fig S3C. Note that the majority of analyses, which use the mixture-model approach are unaffected by food intake data.

To further test whether these 24h food intake measures reliably reflect overall energy intake, we compared the individual measures to cage-based food intake measures collected before the males were separated into individual cages for mating. Six 24h food intake measures of group cages were recorded throughout the duration of group housing to comply with ethics permits. We compared the average of the final two cage-based food intake measures to the average food intake of the 3 individual mice from each cage. Again, we found the intake measures were strongly correlated ($p < 0.001$, $R^2 = 0.596$). This relationship is now shown in Fig S3D.

Note, in investigating this issue we had to go back to the raw data and discovered that we had inadvertently used the second measure of food intake rather than the average of the two measures as intended. We have therefore re-run all GAM-based intake-type analyses with the corrected (average) food intake measures. Importantly, none of our main conclusions have been altered by this re-analysis. Rather, it supports for our conclusions have been strengthened, as increasing precision by using the average of the two measures has increased the number of offspring traits that reach significance in GAM model comparisons. All Figures (Fig 4, Fig S3, Fig S4), Tables (Table 1, Table S3-6), and text (lines 142-156) have been revised and updated with the corrected analyses.

- It is not clear whether food intake for offspring monitored or included in assessments?

Unfortunately, we could not measure food intake in the offspring. The size of the experiment meant that it was not logistically feasible to accurately measure food intake of offspring. We agree that whether paternal diet influences offspring appetite is an interesting question, and we are testing this in other experiments in our research group.

- Mothers were kept on 'standard rodent chow' – definition of diets – chow vs. semi-purified should be aligned with the broader animal nutrition literature.

Thank you for pointing at this. We have revised the manuscript throughout.

- Diet of maternal is understandably fixed, but may limit the ability to interpret alignment between paternal and maternal-specific effects and generalizability of findings to other scenarios (e.g. maternal-paternal dietary congruence, what if both father and mother had a high-fat, low-carb diet)?

We agree that interactions between the maternal and paternal diet are important research questions, particularly if one is aiming to mimic the regimen of a household. However, we think this is far beyond the scope of the current study. The limitations of our study, including our inability to test multiple factors simultaneously is acknowledged in the discussion (lines 231-245).

- One housing temperature utilized, as it is known that temperature affects energy intake,

macronutrient selection and reproductive outcomes, there exist challenges for broad interpretation of findings.

The reviewer raises again an excellent point. However, we believe that addressing this point would constitute a separate study. We have used caution in the interpretation of our results and expressed the limitation of our study in the Discussion, for example lines 256-259 and 261-275.

- Extrapolation of findings for clinical relevance to dietary guidelines premature and incomplete, overreaching.

In our conclusion paragraph, we did not mean to imply that our results can directly inform clinical dietary guidelines, but that was a first step towards providing dietary guidelines. We think indeed that the novel insights gained lay the foundations for future directions in research. However, to avoid any confusion, we have amended the conclusion paragraph (lines 265-269).

Specific Comments

ARRIVE guidelines throughout (e.g. should species be listed in the title, details in figure legends regarding species, sample sizes, etc.?)

Thank you. The manuscript has been checked against the Nature Portfolio Editorial Policy Checklist and Reporting Summary.

Nutritional Geometry framework – capitalization and consistency?

Thank you. This have been amended throughout.

line 35 – Definition of ‘environmental variability’?

Thank you. We have revised this part for more clarity, lines 29-31.

Lines 43-46, not sure this follows...all of the above could share something in common beyond the 'stress' interventions.

We agree with the reviewer that this would be another possibility. Yet, the notion of environmental stress being the common denominator seems the most reasonable explanation to us, but eagerly awaits further studies and alternative hypotheses.

Line 62-64 - Not clear that this is possible within the context of research study utilized here for rodents...

We have amended this text and specified that we refer to mouse as a model (line 62-64).

Line 94 – clarify the ‘adolescence’ stage within the model used for general audience – the ages of animals are noted in the methods and overview figure, but the developmental stage by age of mice may not be known to all readers?

Thank you. We now added the specific age of the animals (lines 90-91).

Line 98-99 and elsewhere – nomenclature/wording... (definitions of ‘high-protein’ ‘low-carbohydrate’, etc. can differ among studies, and although the overall diets are described for macronutrient proportion contents the interpretation of ‘high’ or ‘low’ may not always be self-evident to individual readers).

We agree there is variation in how these terms are applied across studies. Here, we cover the entire physiologically relevant range of macronutrients, and therefore use the terms in a relative sense within the context of our study. It is difficult to describe nutritional patterns observed on the surfaces without uses these terms for grounding.

Line 111 and following (and Line 345 -357) – the collected and presented data regarding ‘behavioral’ outcomes is limited to a single assay and timepoint of measure. The phraseology of ‘metabolic and behavioral traits’ accurately reflects the combined assessment of metabolic outcomes with the single behavioral assessment, but possibly unintentionally results in the reader expecting more than one behavioral assessment to be presented. As other behavioral assays are available and the elevated plus maze reliance on ‘activity’, the presented data offer a specific/narrow assessment of a behavioral trait of the animals. The time of day of measures should also be noted given the sample size utilized and duration of assay performance (it is noted the authors reference publications regarding diurnal phase impact on behavior assays, it is not clear that time of assessment on behavior assay outcomes should be disregarded in analyses to be the consensus in the field).

Thank you for this comment. We found no correlation of time of assay with behavioural traits measured in the EPM, and now include this finding in the results (line 127-129). Furthermore, as the order of testing of animals was randomised, time of day cannot explain the dietary patterns observed in either fathers or male offspring.

We were restricted in the number of behaviour assays that could feasibly be performed in this study due to the number of experimental diets used. We are following up this study with a more thorough investigation of paternal diet effects on a range of offspring behavioural traits, using fewer treatment diets.

Line 169-172 – using individual differences in paternal food intake /calorie consumed – while this was measured, is there any existing knowledge regarding how acute feeding differences can influence epigenetic effects on sperm – e.g. does 1 meal have acute effects or does feeding need to persist for a full day, 1 week, 1 month, etc.? Similarly, if you do not know the amount of intake from the 'brown chow' when paired with females for mating, could this have acute effects on sperm? Would an overview of murine spermatogenic cycle help place the results in context and/or are there windows within the spermatogenic cycle which are more important than others?

The reviewer is raising several excellent questions that need to be investigated in the field of paternal effects. Studies have begun to investigate these more fine-scale effects of timing of feeding on epigenetic changes in sperm and this is an active area of research. Our study investigates longer-term dietary effects, with the dietary intervention spanning at least two spermatogenic cycles (depending on mating success). While it is possible that the brief exposure to chow could have acute effects on sperm, this is an unavoidable experimental constraint that is standard practice in paternal effects studies.

Line 242 – while the presented results are potentially intriguing, it may be a bit presumptuous to consider the work foundational versus more preliminary in light of the overall design and sample sizes, with inherent limitations for systematically modifying multiple factors which would be expected to interact with dietary effects.

This is a reasonable and measured perspective, we meant 'foundational' as being the first step, and have changed the wording accordingly (line 268).

Lines 244-247 - what would the authors propose are the fewer diet comparisons that would be optimal - any attempt to identify the optimal ratio/balances to cover the spectrum sufficiently to address the context-dependence in future studies?

As different traits were found to be influenced by different macronutrients, the subset of diets chosen depends on the trait of interest. For example, if following up effects on female offspring metabolism, diets with a gradient of fat content could be used. In contrast, if following up on effects on male offspring behaviour, diets with a gradient of protein to carbohydrate ratios could be used. We have now added such an example to the discussion line 242-245.

Line 268-269 – 'foundations of preconceptional dietary guidelines for males' – Overreaching. Should be extensively revised or removed?

This has been revised (lines 265-269).

Line 305- what is the specific definition of 'proven ... females'? Were there any differences in pre-study reproductive outcomes for determination of reproductive proven status for the females utilized despite being 'age-matched' to males?

Females that had successfully produced a litter were purchased from ARC. We were not provided with any data about the reproductive outcomes of this previous breeding cycle from the facility. A clarification for what we mean by 'fertility-proven' has been added to the manuscript (line 305-306).

Line 308 - age and exposure differences then – are any of these incorporated in the analyses?

The experiment was not designed or powered to test for effects of these co-variates. We did look for patterns in the data through visual inspection and exploration of basic statistics. No obvious dietary effects were observed in this data exploration.

Line 309 - was litter size adjusted before birth or only after? How separate the programming effects in vivo during pregnancy/development from litter size, etc. versus the paternal diet effects?

Litter size was adjusted 3 days after birth, when sex of pups was clearly identifiable. Handling of dams and litters was minimised directly before and after birth as per ethics requirements.

It is not possible to separate gestational effects from direct paternal effects as the intrauterine environment inevitably contributes to offspring development. However, as the first author has argued in a previous commentary article (Crean & Bonduriansky (2014) *What is a paternal effect?*, Trends Ecol Evol 29: 554-559), we estimate that as the environmental manipulation was performed on the males only, it should be considered a paternal effect, even if the mechanism of action is influenced by the maternal environment.

Line 317-320 – breeding scheme - appears there was food in the cage at night during the mating trial, but not quantified regarding intake and rather stated 'returned to home cages each day to minimize exposure to brown chow', yet most intake occurs during the night – if not quantified, how does one then know or not whether any effect?

As suggested by the reviewer earlier, we cannot rule out that intake during mating may influence offspring health through acute effects. Food intake is not measured during mating events as it is not possible to identify food that is consumed by the male versus the female. It is considered more important to ensure that females are never exposed to treatment diets, so it is common practice in paternal effects studies to place the male into the female cage. All males are exposed to the same diet during mating, so this is a common factor across experimental treatments.

Line 323 - half of the sample for two of the diets failed to produce any offspring from 3 mating cycles? Similarly, was the consideration of 1, 2 or 3 mating cycles or failure to reach reproductive success considered in the analyses for outcomes?

All of the males were found to have viable, motile sperm post cull (reproductive traits of paternal mice reported in companion paper (Crean et al., (2023) *Male reproductive traits are differentially affected by dietary macronutrient balance but unrelated to adiposity*, Nat Comms 14: 2566). Hence, it appears that the failure to produce offspring was a compatibility issue.

Eighty percent of mating pairs produced offspring from the first mating cycle. We visually inspected offspring from pairs that required 2 and 3 mating cycles to produce offspring, and they were dispersed throughout the dataset – no outlier effects were apparent.

Line 329 – based on the light:dark cycle reported, food removal from 9a-1p would likely be even longer period of food deprivation (with 6a:6p light cycle), light phase then during normal sleep/non-feeding state?

This is standard protocol for glucose tolerance tests, widely used in nutrition studies. We maintain consistent protocols across studies for comparative purposes.

Line 338-340 – for the two 24-hour periods of food intake measurement at 16 and 20 weeks of age, does this mean one 24-hr measure within week 16 and one 24-hr measure within week 20 of age or two 24-hr measures performed each of those two weeks (e.g. two 24-hr measures in week 16 and two 24-hr measures in week 20 of age)? Food intake can be quite variable from day-to-day and is sensitive to different types of environmental disturbances (e.g. cage change, outcome measurement interaction, etc.). With the ‘average of both measures’ being used in the analyses (line 342) is there any knowledge of how representative the value was of overall intake behavior?

Please refer to response to similar question from this reviewer above.

A single measure was obtained at each of 16 and 20 weeks of age. As seen in the correlations now provided in Fig S3, although some variability among measures is observed, the broad patterns of increased intake on low protein diets remain consistent throughout.

Lines 386-399 – It appears the diet composition assessments consider the macronutrient composition. It also appears dietary fiber was differential across some dietary macronutrient balances as it appears to be utilized to help maintain dietary energy density. Considering the difference in fiber among groups, was fiber also considered in the analyses of outcomes?

The reviewer raises an important point. While insoluble dietary fibre does not have any significant nutritional value, we appreciate that this component necessarily differed across diets to dilute

calories. It is important to note that, because of the nature of mixtures, it is impossible to break the correlation between either fat and cellulose (in a design such as ours here, which keeps energy density constant), or between fat and energy (in a design that maintains dry mass relationships between macronutrients but therefore allows energy density to vary. We have employed designs which do both in single generation studies (e.g., Solon-Biet et al., (2014) *The ratio of macronutrients, not caloric intake, dictates cardiometabolic health, aging, and longevity in ad libitum-fed mice*, Cell Metab 19: 418-430; Solon-Biet et al., (2015) *Macronutrient balance, reproductive function, and lifespan in aging mice*, PNAS 112: 3481-3486). However, this requires multiplying the number of dietary treatments by each energy density tested, which would be extremely challenging logistically.

Line 428 – Table 1 – how was multiple testing addressed? Red may also not be an optimal shading color given its role in fold and direction changes in other uses (e.g. microarray, etc).

Analyses have not been corrected for multiple testing because: 1) it is not clear whether it is appropriate to consider tests as ‘independent’ and therefore in need of correction when applied to clusters of metabolic/EPM traits. For example, we see effects of diet on fat mass, % fat mass, % lean mass, iAUC, Matsuda index and HOMA, and it seems unlikely anyone of these is a false positive, given these metabolic measures are interrelated, 2) there is not a good body of theoretical literature about whether more stringent AIC values should be used for mixture models when one tests multiple traits. We think it is important for the reviewer to note that false positives are far less likely to occur in the models used in this study compared to standard regressions or ANOVAs.

The shading has been changed to blue.

Table on 430 – what does column 4 (first column under F1 males) represent via the number - what is 5? Does 1-4 represent the models?

Yes, the numbers indicate the mixture model selected by AIC. This has been clarified in the Table description. The models are specified by the authors of the *mixexp* package. Model 5 is referred to as a ‘special cubic’ model.

Line 443 – Fig 2 - why metabolic for female offspring and not male offspring as well? b/e weight of ‘paired’ subcutaneous, c/f weight of paired interscapular – meaning of ‘paired’? How distinguish brown adipose tissue activity from weight of dissected tissue?

Male offspring are not shown because there was no significant relationship with paternal diet (see Table 1) and thus surfaces simply show a flat plane at the overall average (null model).

These fat deposits are located on both sides of the body, in a paired nature. We removed both deposits and measured the combined weight. The wording has been changed in the Figure description for clarity. We show brown adipose tissue weight, not activity. The brown adipose tissue is discernible from the surrounding white adipose tissue during dissection by a difference in colour and texture.

Line 440 - Fig 3 - Why one sex assessed for offspring – rationale for choice?

Both sexes were assessed, but significant effects of paternal diet were only found in male offspring for behavioural traits (summarised in Table 1).

Line 457 – Fig. 4 - why chose the 3rd quartile – how determination made?

The third quartile slice shows data in the region commonly explored in 'high fat' paternal effect studies. Note, the same pattern of nutrient effects is seen in in the 1st and 2nd quartiles (shown in Fig S4), although the range of data is reduced in these slices. We have now directed readers to the other slices in the description of Fig 4.

Suppl Figs

Sup Fig 1 – expand legend and clarify abbreviations used, how adjust for number of attempted matings?

Amended.

Sup Fig 2 – clarify in the figures which animals these represent and time of measure?

Amended.

Sup Fig 3 – Some diets have 50% higher intake among paternal cohort?

Correct. The macronutrient balance of diets has a significant impact on the amount of food consumed. Additional data has been added to Fig S3 as explained above.

Sup Fig 4 – Useful information that could use clarification and context in figure legend?

Amended.

REVIEWERS' COMMENTS

Reviewer #1 (Remarks to the Author):

The authors have done an excellent job responding to this reviewer's concerns. This is a very interesting and timely paper.

Reviewer #2 (Remarks to the Author):

The responsiveness to the previous comments and questions is appreciated. The accompanying revisions have addressed the majority of concerns raised.

One minor concern regarding the wording of the last sentence of the abstract. While the qualifier 'may' is utilized, as written the sentence seems to broadly imply established knowledge regarding the impact of macronutrient balance for optimizing paternal effects on health. While the current study leverages a multi-group assessment of dietary macronutrient balance within the mouse model for paternal effects (which were observed within the constraints of the experimental design utilized), it is not clear whether the statement would be interpreted as making suggestions extending beyond murine species tested or beyond the scope of the findings reported (which in and of themselves may be subject to the range of experimental parameters chosen as noted by the authors in various locations of the manuscript).